# VisualPredicator: Learning Abstract World Models with Neuro-Symbolic Predicates for Robot Planning

**Yichao Liang**[1], **Nishanth Kumar**[3], **Hao Tang**[2], **Adrian Weller**[1,6], **Joshua B. Tenenbaum**[3],

**Tom Silver**[4], **João F. Henriques**[5], **Kevin Ellis**[2]

[1]University of Cambridge, [2]Cornell University, [3]Massachusetts Institute of Technology,
[4]Princeton University, [5]University of Oxford, [6]The Alan Turing Institute

## Abstract

Broadly intelligent agents should form task-specific abstractions that selectively expose the essential elements of a task, while abstracting away the complexity of the raw sensorimotor space. In this work, we present *Neuro-Symbolic Predicates*, a first-order abstraction language that combines the strengths of symbolic and neural knowledge representations. We outline an online algorithm for inventing such predicates and learning abstract world models. We compare our approach to hierarchical reinforcement learning, vision-language model planning, and symbolic predicate invention approaches, on both in- and out-of-distribution tasks across five simulated robotic domains. Results show that our approach offers better sample complexity, stronger out-of-distribution generalization, and improved interpretability.

## 1 Introduction

Planning and model-based decision-making for robotics demands an understanding of the world that is both perceptually and logically rich. For example, a household robot needs to know that slippery objects, such as greasy spatulas, are hard to grasp. Determining if the spatula is greasy is a subtle perceptual problem. As an example of logical richness, for a robot to use a balance beam to weigh objects, it must count up the mass on each side of the balance beam to determine which way the beam will tip. Counting and comparing masses are logically sophisticated operations.

In this work, we show how to efficiently learn symbolic abstractions that are both perceptually and logically rich, and which can plug into standard robot task-planners to solve long-horizon tasks. We consider a robot that encounters a new environment involving novel physical mechanisms and new kinds of objects, and which must learn how to plan in this new environment from relatively few environment interactions (the equivalent of minutes or hours of training experience). The core of our approach is to learn an abstract model of the environment in terms of *Neuro-Symbolic Predicates* (*NSPs*, see Fig. 1), which are snippets of Python code that can invoke vision-language models (VLMs) for querying perceptual properties, and further algorithmically manipulate those properties using Python, in the spirit of ViperGPT and VisProg (Surís et al., 2023; Gupta & Kembhavi, 2022).

In contrast, traditional robot task planning uses hard-coded symbolic world models that cannot adapt to novel environments (Garrett et al., 2021; Konidaris, 2019). Recent works pushed in this direction with limited forms of learning that restrict the allowed perceptual and logical abstractions, and which further require demonstration data instead of having the robot explore on its own (Silver et al., 2023; Konidaris et al., 2018). The representational power of *Neuro-Symbolic Predicates* allows a much broader set of perceptual primitives (essentially anything a VLM can perceive) and also deeper logical structure (in principle, anything computable in Python).

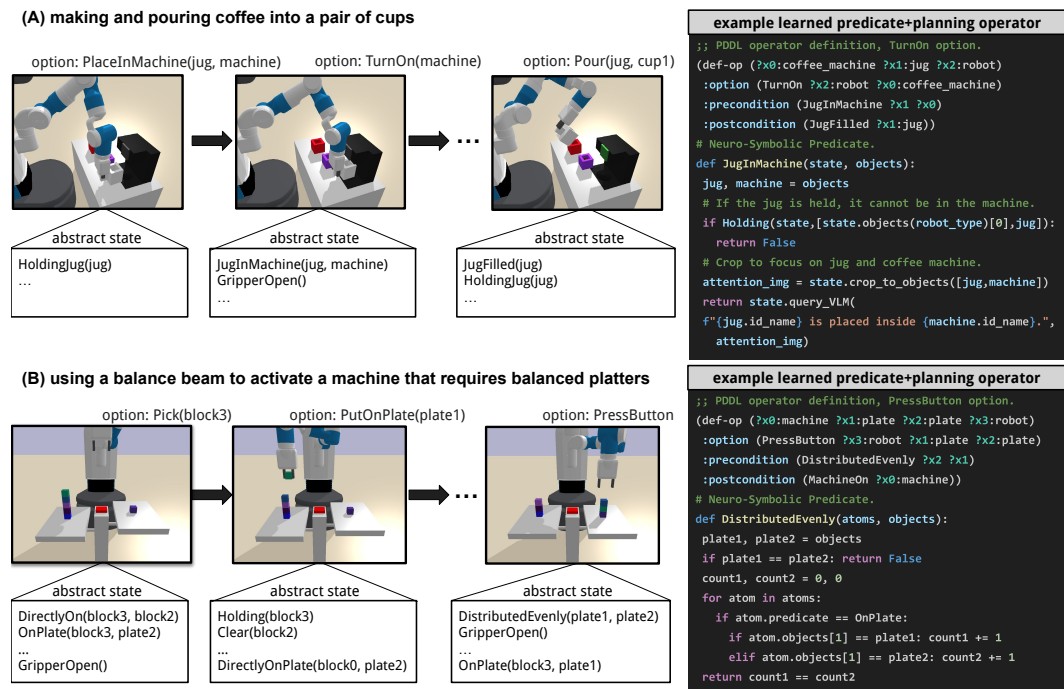

Figure 1: Robot learning domains illustrating learned Neuro-Symbolic predicates. In (A) we learn a predicate that queries a VLM to check if a coffee jug is inside a coffee machine. In (B) we learn a predicate that checks if a balance beam is balanced. (Code lightly refactored to better fit in figure.)

Yet there are steep challenges when learning *Neuro-Symbolic Predicates* to enable effective planning. First, the predicates must be learned from input pixel data, which is extremely complex and potentially noisy. Second, they should not overfit to the situations encountered during training, and instead zero-shot generalize to complex new tasks at test time. Third, we need an efficient way of exploring different possible plans to collect the data needed to learn good predicates. To address these challenges we architect a new robot learning approach that interleaves proposing new predicates (using VLMs), predicate scoring/validation (adapting the modern predicate-learning algorithm by Silver et al. (2022)), and goal-driven exploration with a planner in the loop. The resulting architecture is then able to successfully learn across five different simulated environments, and is more flexible and more sample-efficient compared to competing neural, symbolic, and LLM baselines.

We highlight the following contributions: (1) *NSPs*, a state representation for decision-making using both logically and perceptually rich features; (2) An algorithm for inventing *NSPs* by interacting with an environment, including an extension to a new operator learning algorithm; and (3) Evaluation against 6 methods across 5 simulated robotics tasks.

## 2 PROBLEM FORMULATION

We consider the problem of learning state abstractions for robot planning over continuous state/action spaces, and doing so from online interaction with the environment, rather than learning from human-provided demonstrations. We assume a predefined inventory of basic motor skills, such as pick/place, and also assume a basic object-centric state representation (explained further below), which is a common assumption (Kumar et al., 2024; Silver et al., 2023; 2022). The goal is to learn state abstractions from training tasks that generalize to held-out test tasks, enabling the agent to solve as many test tasks as possible while using minimal planning budget.

**Tasks.** A task $T$ is a tuple $\langle \mathcal{O}, x_0, g \rangle$ of objects $\mathcal{O}$, initial state $x_0$, and goal $g$. The allowed states depend on the objects $\mathcal{O}$, so we write the state space as $\mathcal{X}_{\mathcal{O}}$ (or just $\mathcal{X}$ when the objects are clear from context). Each state $x \in \mathcal{X}_{\mathcal{O}}$ includes a raw RGB image and associated object features, such as 3D object position.

**Environments.** Tasks occur within an environment $\mathcal{E}$, which is a tuple $\langle \mathcal{U}, \mathcal{C}, f, \Lambda \rangle$ where $\mathcal{U} \subseteq \mathbb{R}^m$ is a low-level action space (e.g. motor torques), $\mathcal{C}$ is a set of controllers for low-level skills (e.g. pick/place), $f : \mathcal{X} \times \mathcal{U} \to \mathcal{X}$ is a transition function, and $\Lambda$ is a set of *object types* (possible outputs of an object classifier). The environment is shared across tasks.

**Built-in Motor Skills.** We assume skills $\mathcal{C}$, each of which has parameters that abstract over which object(s) the skill acts on. For example, the agent can apply a skill such as `Place(?block1, ?block2)` to stack any pair of blocks atop one another, where a block is a type in $\Lambda$. We assume the agent can determine whether a skill has been successfully executed upon completion. Skills can be modeled within the options framework (Sutton et al., 1999). The skills $\mathcal{C}$ and the objects $\mathcal{O}$ induce an action space $\mathcal{A}_{\mathcal{O}}$ (or simply $\mathcal{A}$ when the context is clear).

Skills, tasks, and environments are the primary inputs to our system. The primary outputs—what we actually learn—are higher-level abstractions over these basic states and actions.

**Predicates: Abstracting the State.** A predicate $\psi$ is a Boolean feature of a state, which can be parametrized by specific objects in that state. We treat this as function $\psi : \mathcal{O}^m \to (\mathcal{X} \to \mathbb{B})$ that is an indicator, given $m$ objects, of whether a predicate holds in a state. For example, the predicate `On(?block1, ?block2)` inputs a pair of blocks, and outputs a state classifier for whether the first block is atop the second block. A set of predicates $\Psi$ induces an **abstract state** corresponding to all the predicate/object combinations that hold in the current state:

$$\text{ABSTRACT}_\Psi(x) = \{(\psi, o_1, ..., o_m) \; : \; \psi(o_1, ..., o_m) \text{ holds in state } x, \text{ for } \psi \in \Psi \text{ and } o_j \in \mathcal{O}\} \quad (1)$$

We write $\mathcal{S}$ for the set of possible abstract states.

**High-Level Actions: Refining the action space.**[1] Planning requires predicting how each skill transforms the abstract state representation. To make these predictions, High-Level Actions (HLAs) augment skills with *preconditions* specifying which abstract states allow successful use of that skill, and *postconditions*, specifying how the skill transforms the abstract state. Like predicates, an HLA is parametrized by the specific objects it acts upon. Formally, an HLA $\omega$ is a function from a tuple of objects in $\mathcal{O}^m$ to a tuple $\langle \pi, \text{PRE}, \text{EFF}^+, \text{EFF}^- \rangle$ where $\pi \in \mathcal{A}_{\mathcal{O}}$ is a skill, PRE is the precondition, and the postcondition consists of $\text{EFF}^+$ (predicates added to the abstract state) and $\text{EFF}^-$ (predicates removed from the abstract state).

As an example of an HLA, consider `PlaceOnTable(?block, ?table, ?underBlock)`, with PRE = $\{$`Clear(?block)`$\}$, $\text{EFF}^+$ = $\{$`On(?block,?table)`$\}$, and $\text{EFF}^-$ = $\{$`On(?block,?underBlock)`$\}$, using skill $\pi$ = `place(?block,?table)`. This means placing a block on a table, which was previously on top of underBlock, causes the block to be on the table, and no longer on top of underBlock. This HLA is formally a function with arguments `?block,?table,?underBlock`.

**HLA Notation.** We write $\Omega$ for the set of HLAs (what the agent learns), and $\Omega_{\mathcal{O}}$ for their instantiations on objects $\mathcal{O}$ (how the agent uses them in a particular task). We use the variable $\omega$ for HLAs, so we would write $\omega \in \Omega$. We use $\underline{\omega}$ for HLAs applied to particular objects, so we'd write $\underline{\omega} = \langle \underline{\pi}, \underline{\text{PRE}}, \underline{\text{EFF}^+}, \underline{\text{EFF}^-} \rangle \in \Omega_{\mathcal{O}}$.[2]

**Abstract State Transitions.** The predicates and HLAs together define an abstract world model, whose transition function $F : \mathcal{S} \times \Omega_{\mathcal{O}} \to \mathcal{S}$ is

$$F\left(s, = \langle \underline{\pi}, \underline{\text{PRE}}, \underline{\text{EFF}^+}, \underline{\text{EFF}^-} \rangle\right) = \begin{cases} s \cup \underline{\text{EFF}^+} \setminus \underline{\text{EFF}^-} & \text{if } \underline{\text{PRE}} \subseteq s \\ \text{undefined} & \text{otherwise} \end{cases} \quad (2)$$

Having learned predicates and high-level actions, we then solve problems by hierarchical planning:

**A low-level plan** is a sequence of $n$ skills applied to objects $(\underline{\pi_1}, \ldots, \underline{\pi_n}) \in \mathcal{A}_{\mathcal{O}}^n$. It solves a task with goal $g$ and initial state $x_0$ if sequencing those skills starting from $x_0$ satisfies $g$.

**A high-level plan** is a sequence of $n$ HLAs applied to objects, $\underline{\omega}_1, \ldots, \underline{\omega}_n$.

**A note on types.** Because the environment provides object types, we augment predicates and HLAs with typing information for each object-valued argument. Equivalently, predicates return false, and skills terminate immediately with failure, when applied to arguments of the wrong type.

---

[1]In the planning literature, High-Level Actions are also sometimes called operators.

[2]In the planning literature, $\omega$ is called a lifted operator, while $\underline{\omega}$ would be a grounded operator.

# 3 NEURO-SYMBOLIC PREDICATES

*Neuro-Symbolic Predicates* (*NSPs*) represent visually grounded yet logically rich abstractions that enable efficient planning and problem solving. As Figure 2 illustrates, these predicates are neuro-symbolic because they combine programming language constructs (conditionals, numerics, loops and recursion) with API calls to neural vision-language models for evaluating visually-grounded natural language assertions. *NSPs* can be grounded in visual perception, and also in proprioceptive and object-tracking features, such as object poses, common in robotics (Kumar et al., 2024; 2023b; Curtis et al., 2022; 2024b). We consider two classes of NSPs: primitive and derived. Primitive *NSPs* are evaluated directly on the raw state, such as `Holding(obj)` (which would use VLM queries) or `GripperOpen()` (which would use proprioception). Derived *NSPs* instead determine their truth value based on the truth value of other *NSPs*, analogous to derived predicates in planning (Thiébaux et al., 2005; McDermott et al., 1998).

**Primitive *NSPs*.** We provide a Python API for computing over the raw state, including the ability to crop the image to particular objects and query a VLM in natural language. See Appendix A.

**Derived *NSPs*.** Instead of querying the raw state, a derived *NSP* computes its truth value based only on the truth value of other *NSPs*. Derived *NSPs* handle logically rich relations, such as `OnPlate` in fig. 2, which recursively computes if a block is on a plate, or on something that is on a plate.

**Evaluating Primitive *NSPs*.** No VLM is 100% accurate, even for simple queries like "is the robot holding the jug?", especially in partially observable environments. To increase the accuracy and precision of *NSPs*, we take the following two measures (see Appendix B.1 for an example prompt).

First, because a single image may not uniquely identify the state (e.g. due to occlusion), we provide extra context to VLM queries. Consider a robot whose gripper is next to a jug, but whose own arm occludes the jug handle, making it uncertain whether the jug is held by the gripper or merely next to it. Knowing the previous action (e.g. `Pick(jug)`) helps resolve this uncertainty. We therefore further condition *NSPs* on the previous action, as well as the previous visual observation (immediately before the previous action was executed) and previous truth values for the queried ground atom.

Second, we visually label each object in the scene by overlaying a unique ID number on each object in the RGB image (following Yang et al., 2023). That way, to evaluate for example `Holding(block2)`, we can query a VLM with "the robot is holding block2", where block2

```python
def Holding(state: RawState, objects: Sequence[Object]) -> bool:
    """Is the robot holding the block."""
    block, = objects
    # The block can't be held if the robot's hand is open.
    robot = state.get_objects(_robot_type)[0]
    if state.get(robot, "fingers") >= 0.5:
        return False

    attention_image = state.crop_to_objects([block, robot])
    return evaluate_simple_assertion(f"{block.id_name} is held by the robot", attention_image)

def OnPlate(atoms: Set[GroundAtom], objects: Sequence[Object]) -> bool:
    """Whether a block x is directly or transitively on a plate y."""
    block, plate = objects
    for atom in atoms:
        if atom.predicate == DirectlyOnPlate and atom.objects == [block, plate]:
            return True
    other_blocks = {a.objects[0] for a in atoms if a.predicate == DirectlyOn or a.predicate == DirectlyOnPlate}
    for other_block in other_blocks:
        holds1 = False
        for atom in atoms:
            if atom.predicate == DirectlyOn and atom.objects == [block, other_block]:
                holds1 = True
                break
        if holds1 and OnPlate(atoms, [other_block, plate]):
            return True
    return False
```

Figure 2: Example classifiers for `Holding` and `OnPlate` *NSP*.

is labeled with "2." This disambiguates the objects in a scene, allowing an *NSP* to reason precisely about *which* block is held, rather than merely represent that *some* block is held.

**How Derived *NSPs* interact with HLAs.** HLAs form an abstract world model that predicts which predicates are true after performing a skill (the postcondition). Derived predicates do not need to occur in the postcondition, because we can immediately calculate which derived predicates are true based on the predicted truth values of primitive NSPs. Therefore, HLAs can have derived predicates in the precondition, but never in the postcondition.

## 4 HIERARCHICAL PLANNING

We use the learned abstract world model to first make a high-level plan (sequence of HLAs), which then yields a low-level action sequence by calling the corresponding skill policy for each HLA. High-level planning leverages widely-used fast symbolic planners, which, for example, conduct A* search with automatically-derived heuristics (e.g. LM-Cut, Helmert & Domshlak, 2009).

However, there may be a mismatch between a high-level plan, which depends on potentially flawed abstractions, and its actual implementation in the real world. Learning is driven by these failures. More precisely, hierarchical planning can break down in one of two ways:

**Planning Failure #1: Infeasible.** A high-level plan is **infeasible** if one of its constituent skills fails to execute.

**Planning Failure #2: Not satisficing.** A high-level plan is **not satisficing** if its constituent skills successfully execute, but do not achieve the goal.

When solving a task we generate a stream of high-level plans and execute each one until a satisficing plan (achieving the goal) is generated, or until hitting a planning budget $n_{\text{abstract}}$.

## 5 LEARNING AN ABSTRACT WORLD MODEL FROM INTERACTING WITH THE ENVIRONMENT

Algorithm 1 shows how we interleave learning predicates (state abstraction), learning HLAs (abstract transition function), and interacting with the environment. The learner takes in an environment $\mathcal{E}$, a set of training tasks $\mathcal{T}$, an initial predicate set $\Psi_0$ (which is usually the goal predicates), an initial set of HLAs $\Omega_0$ (which are largely empty, section 5.1), and an initial dataset $\mathcal{D}$ (empty, except when doing transfer learning from earlier environments). It tracks its learning progress using $\rho_{\text{best}}$, the highest training solve rate, and $\nu_{\text{best}}$, the lowest number of infeasible plans.

---

**Algorithm 1** Online Pred. Invention($\mathcal{E}, \mathcal{T}, \Psi_0, \Omega_0, \mathcal{D}$)

1: **init:** $\rho_{\text{best}} \leftarrow -\infty$, best solve rate
2: **init:** $\nu_{\text{best}} \leftarrow \infty$, best number of failed plans
3: **init:** $\Psi' \leftarrow \Psi_0$
4: **for** $i \in \text{range}(1, n_{\text{max\_ite}})$ **do**
5: $\quad \mathcal{D}_i, \rho_i, \nu_i \leftarrow \text{Explore}(\Psi_{i-1}, \Omega_{i-1}, \mathcal{E}, \mathcal{T})$ $\triangleright$ section 5.1
6: $\quad$ **if** $\rho_i > \rho_{\text{best}}$ or ($\rho_i = \rho_{\text{best}}$ and $\nu_i < \nu_{\text{best}}$) **then**
7: $\quad\quad \Psi_{\text{best}}, \Omega_{\text{best}}, \rho_{\text{best}}, \nu_{\text{best}} \leftarrow \Psi_i, \Omega_i, \rho_i, \nu_i$
8: $\quad$ **if** $\nu_i = 0$ **then**
9: $\quad\quad$ break
10: $\quad \mathcal{D} \leftarrow \mathcal{D} \cup \mathcal{D}_i$
11: $\quad$ **if** $\rho_i \leq \rho_{i-1}$ or ($\rho_i = \rho_{i-1}$ and $\nu_i > \nu_{i-1}$) **then**
12: $\quad\quad \Psi' \leftarrow \Psi' \cup \text{Propose } NSPs(\mathcal{D}, \Psi_{i-1})$ $\triangleright$ section 5.2
13: $\quad \Psi_i \leftarrow \text{Select Predicates}(\mathcal{D}, \Psi')$ $\triangleright$ section 5.3
14: $\quad \Omega_i \leftarrow \text{Learn HighLevelActions}(\mathcal{D}, \Psi_i)$ $\triangleright$ section 5.4
15: **return** $\Psi_{\text{best}}, \Omega_{\text{best}}$

---

### 5.1 EXPLORATION

Our agent explores the environment by planning with its current predicates/HLAs, and executing the plans. The agent is initialized with underspecified, mostly empty HLA(s) (that is, the preconditions and effects are mostly empty sets, except with goal predicates in the effects if appropriate, so that the planner can generate plans).[3] It collects data by trying to solve the training tasks (generate and execute abstract plans until the task is solved or $n_{\text{abstract}}$ plans are used, as described in section 4) and collects positive transition segments (from successfully-executed skills), negative state-action tuples (from skills that failed to execute successfully) and satisficing plans, if any.

---

[3]Alternatively, it could perform exploration through random skill selection.

## 5.2 PROPOSING PREDICATES

We introduce three strategies for prompting VLMs to invent diverse, task-relevant predicates – two that are conditioned on collected data, and one that is not (see Appendix B.4 for further details).

**Strategy #1 (Discrimination)** helps discover predicates that are good preconditions for the skills. We prompt a VLM with example states where a skill succeeded and failed, and ask it to generate code that predicts when the skill is applicable.

**Strategy #2 (Transition Modeling)** helps discover predicates helpful for postconditions. We prompt a VLM with before (or after) snapshots of successful skill execution, and ask it to generate code that describes properties that changed before (or after, respectively).

**Strategy #3 (Unconditional Generation)** prompts VLMs to propose new predicates as logical extensions of existing ones (whether built-in or previously proposed), without conditioning on the raw planning data. This approach helps create derived predicates.

## 5.3 SELECTING A PREDICATE SET

VLM-generated predicates typically have low precision—not all generations are useful or sensible—and too many predicates will overfit the model to what little data it has collected. One solution could be the propose-then-select paradigm (Silver et al., 2023). Silver et al. (2023) propose an effective predicate selection objective but requires around 50 expert plan demonstrations. We assume *no* demonstration data, and in general, we might not find *any* satisficing plans early in learning. Therefore we need a new way of learning from unsuccessful plans.

To address this, we devise a novel objective that scores a set of predicates $\Psi$ based on classification accuracy, plus a simplicity bias. The objective score is obtained by first learning HLAs using the set of predicates $\Psi$ (discussed more in section 5.4), and then computing the classification accuracy of the HLAs (see Appendix B.3). Later in learning, after discovering *enough* (a hyperparameter one can choose) satisficing plans, we switch to the objective of Silver et al. (2023), which takes planning efficiency and simplicity into account.

We perform a greedy best-first search (GBFS) with either score function as the heuristic. It starts from the set of goal predicates $\Psi_G$ and adds a single new predicate from the proposed candidates at each step, and finally returns the set of predicates with the highest score. This effectively selects $\Psi_i \leftarrow \arg\min_{\Psi \subseteq \mathcal{P}(\Psi)} J(\Psi')$, where $J(\cdot)$ is the objective function. In our experiments, we found the search space is small enough that the enumeration typically takes just a few minutes on a single CPU. For larger search spaces, local hill climbing could be used in place of GBFS.

## 5.4 LEARNING HIGH-LEVEL ACTIONS

We further learn high-level actions $\Omega$, which define an abstract transition model, in the learned predicate space, from interactions with the environment. We follow the *cluster and intersect* operator learning algorithm (Chitnis et al., 2022) and improve its precondition learner for more efficient exploration and better generalization. Chitnis et al. (2022) assume given demonstration trajectories and learns restricted preconditions so that the plans are most similar to the demonstrations. Our agent explores the environment from scratch and does not have demonstration data to follow restrictively. On the other hand, our agent needs more optimistic world models to explore unseen situations to solve the task. Our precondition learner ensures that each data in the transition dataset is modeled by one and only one high-level action and minimizes the syntactic complexity of the HLA to encourage optimistic world models. See Appendix B.2 details.

## 6 EXPERIMENTS

We design our experiments to answer the following questions: **(Q1)** How well does our *NSP* representation and predicate invention approach compare to other state-of-the-art methods, including popular HRL or VLM planning approaches? **(Q2)** How do the abstractions learned by our method perform relative to manually designed abstractions and the abstractions before any learning? **(Q3)** How effective is our *NSP* representation compared to traditional symbolic predicates, where classi-

fiers are based on manually selected object features? **(Q4)** What is the contribution of our extended operator learning algorithm to overall performance?

**Experimental Setup.** We evaluated seven different approaches across five robotic environments simulated using the PyBullet physics engine (Coumans & Bai, 2016). Each result is averaged over five random seeds, and for each seed, we sample 50 test tasks that feature more objects and more complex goals than those encountered during training. The agent is provided with 5 training tasks in the Cover and Coffee environments, 10 tasks in Cover Heavy and Balance, and 20 tasks in Blocks. The planning budget $n_{\text{abstract}}$ is set to 8 for all domains except Coffee, where it is set to 100. The approaches are run on a single CPU except for the MAPLE baseline, which utilizes uses a GPU.

**Environments.** We briefly describe the environments used, including their hand-coded closed-loop controllers, which are shared across all approaches. Additional details can be found in appendix D.

1. **Cover.** The robot is tasked with picking and placing specific blocks to cover designated regions on the table, using Pick and Place skills. Training tasks involve 2 blocks and 2 targets, while test tasks increase the difficulty with 3 blocks and 3 targets.

2. **Blocks.** The robot must construct towers of blocks according to a specified configuration, using Pick, Stack, and PlaceOnTable skills. The agent is trained on tasks involving 3 or 4 blocks and tested on more challenging tasks with 5 or 6 blocks.

3. **Coffee.** The robot is tasked with filling cups with coffee. This involves picking up and placing a jug into a coffee machine, making coffee, and pouring it into the cups. The jug may start at a random rotation, requiring the robot to rotate it before it can be picked up. The environment provides 5 skills: Twist, Pick, Place, TurnMachineOn, and Pour. Training tasks involve filling 1 cup, while test tasks require filling 2 or 3 cups.

4. **Cover Heavy.** This is a variant of Cover with "impossible tasks" which asks the robot to pick and placing white marble blocks that are too heavy for it to pick up. The environment retains the same controllers and number of objects as the standard Cover environment. An impossible task is considered correctly solved if the agent determines that the goal is unreachable with its existing skills (i.e., no feasible plan can be generated).

5. **Balance.** In this environment, the agent is tasked with turning on a machine by pressing a button in front of it, but without prior knowledge of the mechanism required to activate it (in this case, balancing an equal number of blocks on both sides). The agent has access to a PressButton skill, along with the skills from the Blocks domain. Training tasks involve 2 or 4 blocks, while test tasks increase the difficulty with 4 or 6 blocks.

**Approaches.** We compare our approach against 5 baselines and manually designed state abstraction.

1. **Ours.** Our main approach.

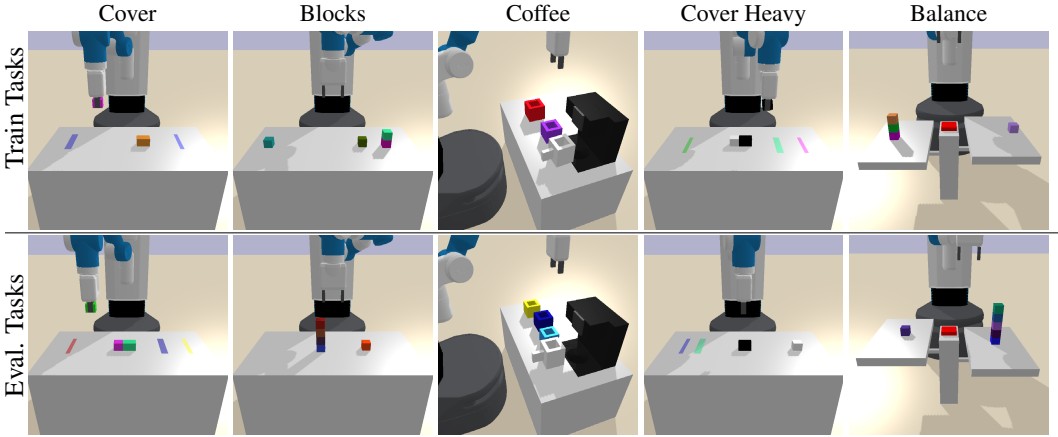

Figure 3: Environments. Top row: train task examples. Bottom row: evaluation task examples.

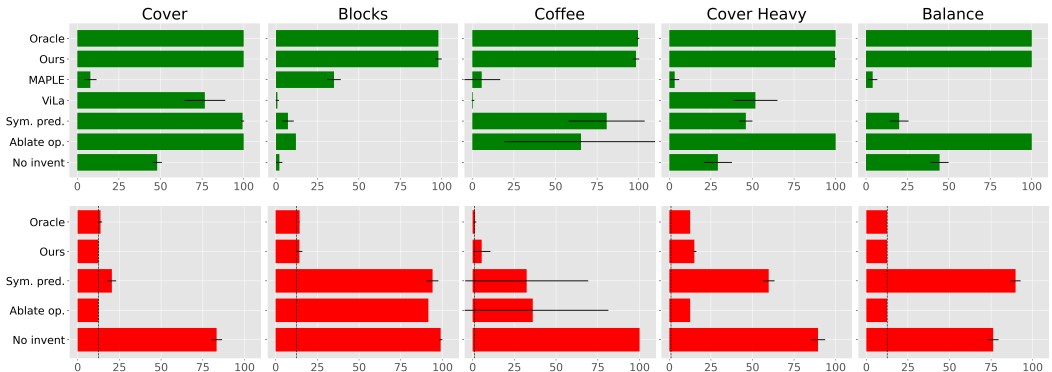

Figure 4: Top row: percentage solved for different domains (↑). Bottom: percentage of planning budget used to find the satisficing plans (↓). The dashed line shows the minimal number of plans needed to solve all the tasks (1 plan per task).

2. **MAPLE.** a HRL baseline that learns to select high-level action by learning a Q function, but does not explicit learn predicates and perform planning. This is inspired by the recent work on MAPLE (Nasiriany et al., 2022b). While we have extended the original work with the capacity of goal-conditioning, the implementation is still not able to deal with goals involving more objects than it has seen during training. Hence, we are only evaluating this approach with tasks from the training distribution.

3. **ViLa** (Hu et al., 2023). A VLM planning baseline which zero-shot prompts a VLM to plan a sequence of actions, without learning.

4. **Sym. pred.** A baseline that uses the same online learning algorithm but only has access to object features that are commonly present in robotics tasks when writing predicates, i.e., without open-ended VLM queries and derived predicates. This shares a similar representation as recent work Interpret (Han et al., 2024) but is still distinct since they mostly learn from human instruction.

5. **Ablate op.** An ablation that does not use our extension to the operator learner.

6. **No invent.** A baseline that uses the abstractions our approach is initialized with and does not perform any learning.

7. **Oracle.** An "oracle" planning agent with manually designed predicates and operators.

**Results and Discussion.** Figure 4 presents the evaluation task solve rate and the planning budget utilized. Examples of an online learning trajectory with invented predicates, instances of learned abstractions, and further planning statistics (such as node expanded and walltime) are provided in appendix C.

Our approach consistently outperforms the HRL and VLM planning baselines, **MAPLE** and **ViLa**, across all tested domains, achieving near-perfect solve rates (**Q1**). With similar amounts of inter-action data, MAPLE struggles to perform well, even on tasks within the training distribution. This limitation could potentially be mitigated with significantly larger datasets, though this is often impractical in robotics due to the high cost of real-world interaction data and the sim-to-real gap in transferring simulation-trained policies. ViLa demonstrates limited planning capabilities, which is consistent with recent observations (Kambhampati et al., 2024). While it performs adequately on simple tasks like Cover, where the robot picks and places blocks, its performance drops significantly when blocks are initialized in the robot's grasp, as it tends to redundantly attempt picking actions. This behavior suggests overfitting. In more complex domains, ViLa often generates infeasible plans, such as attempting to pick blocks from a stack's middle or trying to grasp a jug without considering its orientation. We think introducing demonstrations or incorporating environment interactions could potentially alleviate these issues.

Our approach significantly outperforms **No invent**, demonstrating the clear benefits of learning predicate abstractions over relying on initial underspecified representations. It achieves similar solve rates and efficiency to the **Oracle** baseline, which uses manually designed abstractions (**Q2**). This

underscores the ability of our method to autonomously discover abstractions as effective as those crafted by human experts.

Addressing (**Q3**), while **Sym. pred.** performs well in simple domains like Cover, it struggles to invent predicates that require grounding in perceptual cues not explicitly encoded in object features. For instance, in Coffee, it cannot reliably determine if a jug is inside a coffee machine based on object poses—a key precondition for the `TurnMachineOn` action. Similarly, in Cover Heavy, it fails to recognize blocks that are too heavy to lift, which is critical for identifying unreachable goals. Additionally, without derived *NSPs*, reasoning accurately and efficiently about abstract concepts in the abstract world model (such as whether the number of blocks on both sides of a balance is equal) becomes challenging, which is critical for solving Balance More generally, we hypothesize that providing all feature-value pairs for every object in each state during prompting overwhelms existing VLMs, leading to poor predicate invention. This likely accounts for the subpar performance, even in simple domains like Blocks. These limitations emphasize the strengths of our *NSP* representation and learning pipeline.

Finally, to answer (**Q4**), we find that our approach performs better than **Ablate op.**, which sometimes learns unnecessarily complex preconditions that overfit early, limited data, hindering further learning on training tasks. In other cases, overly specific preconditions result in good training performance but poor generalization, such as requiring `JugInMachine` for the `Pour` action. This demonstrates the value of our operator learner, especially in data-scarce, exploration-based learning settings.

## 7 RELATED WORKS

**Hierarchical Reinforcement Learning** (HRL) HRL tackles the challenge of solving MDPs with high-dimensional state and action spaces, common in robotics, by leveraging temporally extended, high-level actions (Barto & Mahadevan, 2003). The Parameterized Action MDPs (PAMDPs) framework (Masson et al., 2016) builds on this by integrating discrete actions with continuous parameters, optimizing both the action and its parameterization using the Q-PAMDP algorithm. MAPLE (Nasiriany et al., 2022a) further builds on this by using a library of behavior primitives, such as grasping and pushing, combined with a high-level policy that selects and parameterize these actions. We implement a version of this with the extension of goal-conditioned high-level policy as a baseline. Generative Skill Chaining (GSC) (Mishra et al., 2023) further improves long-horizon planning by using skill-centric diffusion models that chain together skills while enforcing geometric constrains. Despite these advancements, they still face challenges in sample complexity, generalization, and interpretability.

**Large Pre-Trained Models for Robotics** With the rise of large (vision) language models (VLMs), many works explore their application in robotic decision making. RT-2 (Brohan et al., 2023) treats robotic actions as utterances in an "action language" learned from web-scale datasets. SayCan and Inner Monologue (Ahn et al., 2022; Huang et al., 2022) use LLMs to select skills from a pretrained library based on task prompts and prior actions. Code as Policy (Liang et al., 2023) prompts LLMs to write policy code that handles perception and control. Recent works extend this to bilevel planning (Curtis et al., 2024a), but do not learn new predicates. ViLa (Hu et al., 2023) queries VLMs for action plans, executing the first step before replanning. We implement an open-loop version of ViLa to compare with its initial planning capabilities.

**Learning Abstraction for Planning** Our work builds on a rich body of research focused on learning abstractions for planning. Many prior works have explored offline methods such as learning action operators and transition models from demonstrations using existing predicates (Silver et al., 2021; Chitnis et al., 2022; Pasula et al., 2007; Silver et al., 2022; Kumar et al., 2023a). While Silver et al. (2023) explore learning predicates grounded in object-centric features, our approach goes further by inventing open-ended, visually and logically rich concepts, without relying on hand-selected features. Additionally, unlike their demonstration-based approach, ours learns purely online. Konidaris et al. (2018) and consequent works (James et al., 2022; 2020) discover abstraction in an online fashion by leveraging the initiable and terminations set of operators that satisfy an abstract subgoal property. James et al. (2020) incorporate an egocentric observation space to learn more portable representations, and James et al. (2022) define equivalence of options effects on objects to derive object types for better transferability. Nevertheless, they work on a constrained class of classifiers (such as decision trees or linear regression with feature selection), which limits the effectiveness and

generalizability of learned predicates. Kumar et al. (2024) performs efficient online learning, but focuses on sampler learning rather than predicate invention.

# 8 Conclusion

In this work, we introduced *Neuro-Symbolic Predicates* (*NSPs*), a novel representation that combines the flexibility of neural networks to represent open-ended, visually grounded concepts, and the interpretability and compositionality of symbolic representations, for planning. To support this, we developed an online algorithm for inventing *NSPs* and learning abstract world models, which allows efficient acquisition of *NSPs*. Our experiments across five simulated robotic domains demonstrated that our method outperforms existing approaches, including hierarchical reinforcement learning, VLM planning, and traditional symbolic predicates, particularly in terms of sample efficiency, generalization, and interpretability. Exciting areas for future work include incorporating recovery mechanisms for failed plans, enhancing exploration efficiency, scaling to partially observable and real-world domains, and relaxing assumptions about skills leverage advances in policy synthesis (Liang et al., 2023), RL (Liang et al., 2024; Ma et al., 2023), or motion planning (Huang et al., 2024).

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

CONTENTS

## A    PYTHON API FOR *NSPs*

We provide the following Python API on for writing primitive *NSPs*: `get_object(t: Type)` returns all objects in the state of a type `t`. `get(o: Object, f: str)` retrieves the feature with name `f` for object `o`. We also have `crop_to_objects(os: Sequence[Object], ...)` for cropping the state observation image to include just the specified list of objects to reduce the complexity for downstream visual reasoning. Finally, there is `evaluate_simple_assertion(a: str, i: Image)` for evaluating the natural language assertion `a` in the context of image `i` using a VLM.

## B    ADDITIONAL DETAILS ABOUT THE ONLINE INVENTION ALGORITHM

### B.1    PREDICATE INTERPRETATION

We provide an example prompt used to interpret the truth value of the ground atom `DirectlyOn(block5, block6)` in the state with cropped observation shown in Figure 5. The highlighted text illustrates how we condition on previous action, previous observation, previous truth value, and object IDs to improve the predicate evaluation accuracy.

> Evaluate the truth value of the following assertions in the current state as depicted by the image labeled with 'curr. state'. For context, the state is right after the robot has successfully executed the action Pick(robot1:robot, block5:block) . The state before executing the action is depicted by the image labeled with 'prev. state'. Please carefully examine the images depicting the 'prev. state' and 'curr. state' before making a judgment. The assertions to evaluate are:
> 1. block5 is directly on top of block6 . (which was False before the successful execution of the previous action)

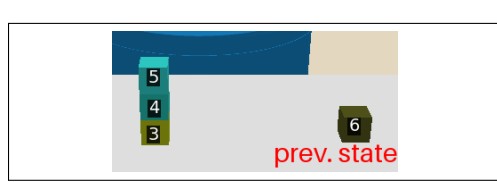
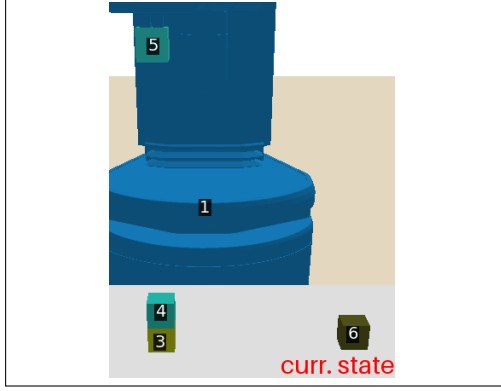

Figure 5: Example cropped current (right) and previous (left) observations used for interpreting ground predicates.

### B.2    LEARNING HLAS BY EXTENDING THE *cluster and intersect* ALGORITHM

We aim to learn high-level actions $\Omega$, which define an abstract transition model in the learned predicate space, from interactions with the environment. These interactions consist of executing high-level plans, which are sequences of (grounded) HLAs $\underline{\omega}_1, \ldots, \underline{\omega}_n$ (i.e. HLAs applied to concrete objects). Our learned abstract transition model should both fit the transition dataset while being optimistic for efficient exploration (Tang et al., 2024). Recalling the definitions from sec. 2, given the current transition dataset, $\mathcal{D} = \{\ldots, (x^{(k)}, \pi^{(k)}, x_\pi^{(k)}), \ldots, (x^{(k')}, \pi^{(k')}, \text{FAIL}), \ldots\}$, we first transform it into the learned abstract state space, $\mathcal{D}_\Psi = \{\ldots, (s^{(k)}, \pi^{(k)}, s_\pi^{(k)})), \ldots, (s^{(k')}, \pi^{(k')}, \text{FAIL}), \ldots\}$, where $s =$

ABSTRACT$_\Psi(x)$. We aim to learn high-level actions, $\Omega$, such that for all high-level actions $\underline{\omega} \in \Omega_{\mathcal{O}}$ on objects $\mathcal{O}$,

$$\forall(s^{(k)}, \pi^{(k)}, s^{(k)}_\pi) \in \mathcal{D}_\Psi, \exists \underline{\omega} \in \Omega^{\pi^{(k)}}_{\mathcal{O}}, \ \underline{\omega}.\text{PRE} \subseteq s^{(k)} \wedge$$
$$s^{(k)}_\pi - s^{(k)} = \underline{\omega}.\text{EFF}^+ \wedge s^{(k)} - s^{(k)}_\pi = \underline{\omega}.\text{EFF}^-,$$
$$\forall(s^{(k)}, \pi^{(k)}, s^{(k)}_\pi) \in \mathcal{D}_\Psi, \forall \underline{\omega} \in \Omega^{\pi^{(k)}}_{\mathcal{O}}, \ \underline{\omega}.\text{PRE} \subseteq s^{(k)} \Rightarrow$$
$$\left( s^{(k)}_\pi - s^{(k)} = \underline{\omega}.\text{EFF}^+ \wedge s^{(k)} - s^{(k)}_\pi = \underline{\omega}.\text{EFF}^- \right),$$
$$\forall(s^{(k)}, \pi^{(k)}, \text{FAIL}) \in \mathcal{D}_\Psi, \nexists \underline{\omega} \in \Omega^{\pi^{(k)}}_{\mathcal{O}}, \ \underline{\omega}.\text{PRE} \subseteq s^{(k)},$$
$$\text{where } \Omega^{\pi^{(k)}}_{\mathcal{O}} = \{\underline{\omega} : \underline{\omega} \in \Omega_{\mathcal{O}} \wedge \underline{\omega}.\pi = \pi^{(k)}\}, \tag{3}$$

while minimizing the syntactic complexity of the HLA, $|\underline{\omega}.\text{PRE}| + |\underline{\omega}.\text{EFF}^+| + |\underline{\omega}.\text{EFF}^-|$.

To find the high-level actions satisfying this objective, we first split the dataset according to the skills, as each high-level action is only associated with one skill, $\mathcal{D}^{\pi_i}_\Psi = \{d : d \in \mathcal{D}_\Psi \wedge d.\pi = \pi_i\}$. We then split each skill into one or multiple high-level actions by unifying the effects in $\mathcal{D}^{\pi_i}_\Psi$ following the *cluster and intersect* operator learner (Chitnis et al., 2022). This compensates for the fact that a skill can have different effects in different situations, by first partitioning the transition datasets into high-level actions,

$$\mathcal{D}^\omega_\Psi = \{d : d \in \mathcal{D}_\Psi \wedge d.\pi = \omega.\pi \wedge d.s^{(k)}_\pi - d.s^{(k)} = \underline{\omega}.\text{EFF}^+ \wedge d.s^{(k)} - d.s^{(k)}_\pi = \underline{\omega}.\text{EFF}^-$$
$$\text{where } \underline{\omega} = \omega(o_1, o_2, \ldots), \text{ for all } o_i \in \mathcal{O}\}.$$

Each partition associates a high-level action with the skill $\omega.\pi = d.\pi, \forall d \in \mathcal{D}^\omega_\Psi$, while the postconditions of the high-level action ($\omega.\text{EFF}^+$ and $\omega.\text{EFF}^-$) are also learned, by unifying and lifting the effects of data in $\mathcal{D}^\omega_\Psi$. See Chitnis et al. (2022) for more details. For the preconditions, $\omega.\text{PRE}$, we learn them by maximizing

$$J(\omega.\text{PRE}) =$$
$$\frac{1}{|\mathcal{D}^{\omega.\pi}_\Psi|} \left( \sum_{d \in \mathcal{D}^\omega_\Psi} \mathbb{1}\left(\omega.\text{PRE} \subseteq d.s^{(k)}\right) + \sum_{d \in \left(\mathcal{D}^{\omega.\pi}_\Psi - \mathcal{D}^\omega_\Psi\right)} \mathbb{1}\left(\omega.\text{PRE} \not\subseteq d.s^{(k)}\right) \right) + \alpha \cdot |\omega.\text{PRE}|. \tag{4}$$

This ensures that all data in the partition is modeled by the associated high-level action, $\omega$. It specifies that the skill $\omega.\pi$ is applicable to states $s^{(k)}$ as $\underline{\omega}.\text{PRE} \subseteq s^{(k)}$. This high-level action also models all other data in the transition dataset, specifying that its precondition is not satisfied if a skill is not applicable on a state, $(s^{(k)}, \omega.\pi, \text{FAIL}) \in \mathcal{D}^{\omega.\pi}_\Psi$, or if a skill has different effects when applied on the state, $(s^{(k)}, \omega.\pi, s^{(k)}_\pi) \in \mathcal{D}^{\omega.\pi}_\Psi \wedge (s^{(k)}, \omega.\pi, s^{(k)}_\pi) \notin \mathcal{D}^\omega_\Psi$. We set the parameter $\alpha$ to a small number, which softly penalizes syntactically complex preconditions.

Compared with the *cluster and intersect* operator learner (Chitnis et al., 2022), which simply intersecting over feasible states to build preconditions for each high-level action, our method optimistically enlarges the set of feasible states for each high-level actions using the minimum complexity objective, while still retaining the abilities to distinguish infeasible states. The optimistic objective is critical for predicate invention by interactions where optimal demonstration trajectories are not available. Using the intersection method, the agent will only consider the feasible states in the currently curated dataset as feasible and never try the skill in other states that are potentially feasible as well. Planners usually fail to find plans with such restrictive world models, resulting in inefficient random exploration and poor test-time performance.

The restricted preconditions are less generalizable as well. For example, for agents learning making coffee in environments with one cup, the agent will find successful trajectories such as PutKettleIn-CoffeeMachine, MakeCoffee, and PourCoffeeInCup. Using the intersection method, the agent sets the preconditions of PourCoffeeInCup as KettleInMachine and KettleHasCoffee as both of them are always true among feasible states of the PourCoffeeInCup action, even though only KettleHasCoffee

is needed. The more restricted preconditions are problematic when generalizing to environments with more than one cups. The agent keeps putting the kettle back to the machine before pouring the coffee for another cup, as the learned PourCoffeeInCup action has KettleInMachine as part of its precondition. The agent eventually fails to solve the problem as the number of cups increases due to the almost doubled length of feasible plans in the more restricted abstract world model. Our method finds the correct precondition as KettleHasCoffee with the optimistic objective. We prefer KettleHasCoffee over KettleInMachine as it fails to distinguish infeasible states for the Pour skill with different effects, PourNothingInCup.

In terms of time complexity, *cluster and intersect* is linear in the number of successful transitions in $\mathcal{D}$ and the number of predicates $\Psi$, where the additional greedy best-first search (GBFS) that we do introduces exponential complexity with respect to the number of predicates. To balance computational efficiency and performance, we use *cluster and intersect* in the inner loop of predicate section and then apply our method to the selected predicates (which is usually less than a dozen). Additionally, local hill climbing can be used as an alternative to GBFS to further improve computational efficiency.

## B.3 CLASSIFICATION-ACCURACY-BASED PREDICATE SETS SCORE FUNCTION

When no satisficing plan is found in early iterations of predicate invention (e.g., in Coffee), the objective from Silver et al. (2023) is inapplicable. This issue is particularly prominent when the space of possible plans is large (i.e., when there are many potential actions at each step and achieving goals requires long-horizon plans). To address this, we introduce a predicate score function that does not rely on satisficing plans. We propose an alternative objective based on classification accuracy, in the same flavour as the score function defined earlier for operator preconditions.

Formally, given $\mathcal{D}_{\Psi} = \{\ldots, (s^{(k)}, \pi^{(k)}, s_{\pi}^{(k)})), \ldots, (s^{(k')}, \pi^{(k')}, \text{FAIL}), \ldots\}$, where $s = \text{ABSTRACT}_{\Psi}(x)$ as above, we denote the collection of all success transitions and failed tuples as $\mathcal{D}_{\Psi}^{+} = \{(s^{(k)}, \pi^{(k)}, s_{\pi}^{(k)}))\}$ and $\mathcal{D}_{\Psi}^{-} = \{(s^{(k)}, \pi^{(k)}, \text{FAIL})$ respectively. The the predicate set score function is

$$J(\Psi) = \frac{1}{|\mathcal{D}_{\Psi}|} \left( \sum_{(s^{(k)}, \pi^{(k)}, s_{\pi}^{(k)}) \in \mathcal{D}_{\Psi}^{+}} \mathbb{1} \left( \exists \omega . \pi = \pi^{(k)} . \omega . \text{PRE} \subseteq s \right) \right.$$

$$\left. + \sum_{(s^{(k)}, \pi^{(k)}, \text{FAIL}) \in \mathcal{D}_{\Psi}^{-}} \mathbb{1} \left( \nexists \omega . \pi^{(k)} = \pi . \omega . \text{PRE} \subseteq s \right) \right) + \alpha \cdot |\Psi|. \tag{5}$$

Intuitively, this objective selects for the minimal set of predicates $\Psi$ such that the HLAs learned from these predicates, $\Omega_{\Psi}$, avoid attempting to execute a skill in states where it has previously failed while ensuring that the HLAs enable the skill to be executed in states where it has previously succeeded.

## B.4 PROMPTING FOR PREDICATES

**Strategy #1 (Discrimination).** is motivated by one of the primary functions of predicates–have them in the preconditions of operators to distinguishing between the positive and negative states so the plans the agent find are feasible. However, we observed that existing VLMs often struggle to reliably understand and identify the differences between positive and negative states, especially when dealing with scene images that deviate significantly from those seen during training. This limitation motivates our second strategy.

**Strategy #2 (Transition Modeling).** With the observation that predicates present in an action's preconditions often also appear in some actions' effects. We prompt the VLM to propose predicates that describe these effects based on the *positive transition segments* it collects. This task is usually easier for VLMs because it involves identifying the properties or relationships that have changed from the start state to the end state, given the information that an action with a natural language name (such as pick) has been successfully executed. However, this strategy alone is not exhaustive. Certain predicates may exist solely within preconditions but not effects (e.g., an object's material that remains unchanged). Therefore, this method complements S1 and is used alternately with it during the invention iterations.

**Strategy #3 (Unconditional Generation)**. prompts VLMs to propose derivations based on existing predicates. These derivations can incorporate a variety of logical operations, such as negation, universal quantification (e.g., defining `Clear(x)` based on `On(x,y)`), transitive closure, and disjunction (e.g., defining `OnPlate(x,p)` based on `DirectlyOn(x,y)` and `DirectlyOnPlate(x,p)`). This approach helps create derived predicates, such as `OnPlate` for `Balanced` (fig. 1). , which is unlikely to be proposed by the first two strategies but are essential for correctly implementing complex predicates like `Balanced`. As a result, this S3 is used at every invention iteration before either S1 or S2 is executed.

For each predicate proposal strategy, we propose a three-step method to guide the VLMs: 1) Ask the VLM to propose predicates by providing a predicate name, a list of predicate types drawn from $\Lambda$, and a natural language description of the assertion the predicate corresponds to. 2) Synthesize the predicates classifiers using the syntax and API we provide for *NSPs* 3) Identify any potential derived predicates and prompt a language model to transform them into the specified function signature for derived *NSPs*. Given the challenges in S1, we add an additional step 0 just for this strategy. We query the VLM to propose properties or relations among objects in natural language, which are then formalized into predicates in Step 1.

### B.5    LIMITATIONS AND FAILURE CASES

A primary limitation of the system is the accuracy and reliability of the VLM in evaluating NSPs.

In some cases, the system can recover from imperfect predicate evaluation accuracies. This is because noisy predicates are not selected during the predicate selection process, and variations of the predicates, with slightly different natural language descriptions, can be proposed in later invention iterations. These variations may achieve higher scores, making them more likely to be selected.

In other cases, they never recover. For instance, in the Cover Heavy domain, our initial plan was to assign common materials, such as wood and metal, to blocks to distinguish between light and heavy objects. While the predicate proposal VLM successfully suggested appropriate predicates (e.g., `IsWood(?block)` and `IsMetal(?block)`), the predicate evaluation VLM was unable to interpret these predicates with sufficient accuracy and consistency to build a useful world model. The issue persisted even after switching to white and black blocks to represent heavy and light blocks and was ultimately resolved by using green and red blocks instead. Similarly, in the Coffee domain, the predicate `IsJugFilled(?jug)` is an essential precondition for the `pour` HLA. However, the VLM could not interpret this predicate accurately enough, necessitating that we treat it as a predefined predicate.

Potential solutions include: 1) integrating proprioception more effectively into the system; 2) developing ways to accurately assign belief scores over the truth values (e.g., use "IsJugFilled(?jug)–0.9" to denote "I believe the jug is filled with coffee with probability 0.9"); Or designing embeddings for ground predicates and observations, and determining the truth values of ground predicates by comparing distances between the corresponding embeddings.

At the same time, we expect improved accuracy in real-world scenarios compared to simulated domains with poor graphics quality, as there should be less distribution shift relative to the VLMs' training data, and the VLMs have demonstrated very strong performance on simple visual question-answering tasks with natural images Yang et al. (2023).

## C    ADDITIONAL EXPERIMENTAL RESULTS

### C.1    EXAMPLE ONLINE LEARNING TRAJECTORY

Figure 6 shows an example of a predicate invention curve in the Coffee environment. Learning begins with 800 failed plans (i.e., unable to solve any tasks) and concludes after 8 iterations when the number of failed plans reaches zero. In total, 9 predicates are selected from 46 candidates. In the end, it selects 9 predicates among 46 candidates.

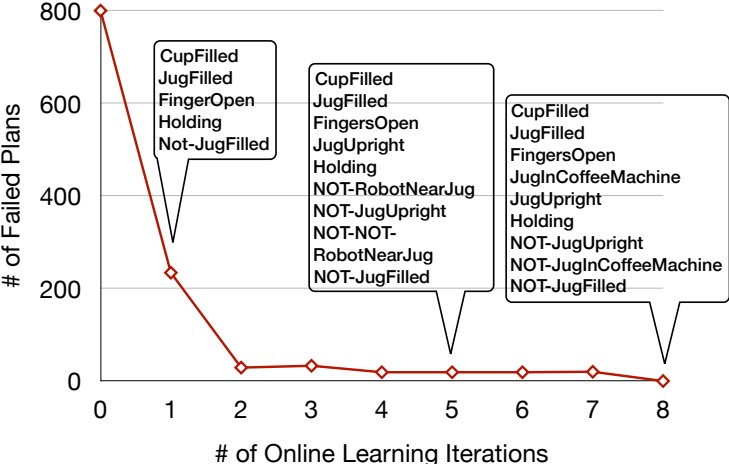

Figure 6: An example online predicate invention trajectory. The bubbles show the predicates being selected among all the candidates it has at that iteration.

## C.2 LEARNED ABSTRACTIONS

We show examples of learned predicates and operators here.

### C.2.1 COVER

```python
def _GripperOpen_NSP_holds(state: RawState, objects: Sequence[Object]) -> bool:
    robot, = objects
    return state.get(robot, "fingers") > 0.5

name: str = "GripperOpen"
param_types: Sequence[Type] = [_robot_type]
GripperOpen = NSPredicate(name, param_types, _GripperOpen_NSP_holds)
```

```python
def _Holding_NSP_holds(state: RawState, objects: Sequence[Object]) -> bool:
    robot, block = objects
    # If the gripper is open, the robot cannot be holding anything
    if state.get(robot, "fingers") > 0.5:
        return False

    # Crop the image to focus on the robot and block
    attention_image = state.crop_to_objects([robot, block])
    robot_name = robot.id_name
    block_name = block.id_name
    return state.evaluate_simple_assertion(
        f"{robot_name} is holding {block_name}", attention_image
    )

name: str = "Holding"
param_types: Sequence[Type] = [_robot_type, _block_type]
Holding = NSPredicate(name, param_types, _Holding_NSP_holds)
```

```
NSRT-Op0:
    Parameters: [?x0:block, ?x1:robot]
    Preconditions: [GripperOpen(?x1:robot)]
    Add Effects: [Holding(?x1:robot, ?x0:block)]
    Delete Effects: [GripperOpen(?x1:robot)]
    Ignore Effects: []
    Option Spec: Pick(?x0:block)
NSRT-Op1:
    Parameters: [?x0:block, ?x1:robot, ?x2:target]
    Preconditions: [Holding(?x1:robot, ?x0:block)]
    Add Effects: [Covers(?x0:block, ?x2:target), GripperOpen(?x1:robot)]
    Delete Effects: [Holding(?x1:robot, ?x0:block)]
    Ignore Effects: []
    Option Spec: Place(?x0:block, ?x2:target)
```

### C.2.2 BLOCKS

```python
1   Gripping
2   ```python
3   def _Gripping_NSP_holds(state: RawState, objects: Sequence[Object]) -> bool:
4       """Determine if the robot in objects is gripping the block in objects
5       in the scene image."""
6       robot, block = objects
7       robot_name = robot.id_name
8       block_name = block.id_name
9
10      # If the robot's fingers are open, it can't be gripping anything.
11      if state.get(robot, "fingers") > 0:
12          return False
13
14      # Crop the scene image to the smallest bounding box that include both objects.
15      attention_image = state.crop_to_objects([robot, block])
16      return state.evaluate_simple_assertion(
17          f"{robot_name} is gripping {block_name}.", attention_image)
18
19  name: str = "Gripping"
20  param_types: Sequence[Type] = [_robot_type, _block_type]
21  Gripping = NSPredicate(name, param_types, _Gripping_NSP_holds)
22  ```
23
24  Clear
25  ```python
26  # Define the classifier function
27  def _Clear_CP_holds(atoms: Set[GroundAtom], objects: Sequence[Object]) -> bool:
28      """Determine if there is no block on top of the given block."""
29
30      block, = objects
31
32      # Check if any block is on top of the given block
33      for atom in atoms:
34          if atom.predicate == On and atom.objects[1] == block:
35              return False
36      return True
37
38  # Define the predicate name here
39  name: str = "Clear"
40
41  # A list of object-type variables for the predicate, using the ones defined in the environment
42  param_types: Sequence[Type] = [_block_type]
43
44  # Instantiate the predicate
45  Clear = ConceptPredicate(name, param_types, _Clear_CP_holds)
46  ```
47
48  EmptyGripper
49  ```python
50  def _EmptyGripper_NSP_holds(state: RawState, objects: Sequence[Object]) -> bool:
51      """Determine if the gripper of robot in objects is empty in the scene image."""
52      robot, = objects
53      # If the robot's fingers are closed, it can't be empty.
54      if state.get(robot, "fingers") < 1:
55          return False
56      return True
```

```
57
58  name: str = "EmptyGripper"
59  param_types: Sequence[Type] = [_robot_type]
60  EmptyGripper = NSPredicate(name, param_types, _EmptyGripper_NSP_holds)
61  ```
```

```
NSRT-Op0:
    Parameters: [?x0:block, ?x1:block, ?x2:robot]
    Preconditions: [Clear(?x1:block), EmptyGripper(?x2:robot),
    On(?x1:block, ?x0:block)]
    Add Effects: [Gripping(?x2:robot, ?x1:block)]
    Delete Effects: [EmptyGripper(?x2:robot), On(?x1:block, ?x0:block)]
    Ignore Effects: []
    Option Spec: Pick(?x2:robot, ?x1:block)
NSRT-Op1:
    Parameters: [?x0:block, ?x1:robot]
    Preconditions: [Gripping(?x1:robot, ?x0:block)]
    Add Effects: [EmptyGripper(?x1:robot), OnTable(?x0:block)]
    Delete Effects: [Gripping(?x1:robot, ?x0:block)]
    Ignore Effects: []
    Option Spec: PutOnTable(?x1:robot)
NSRT-Op2:
    Parameters: [?x0:block, ?x1:robot]
    Preconditions: [Clear(?x0:block), EmptyGripper(?x1:robot),
    OnTable(?x0:block)]
    Add Effects: [Gripping(?x1:robot, ?x0:block)]
    Delete Effects: [EmptyGripper(?x1:robot), OnTable(?x0:block)]
    Ignore Effects: []
    Option Spec: Pick(?x1:robot, ?x0:block)
NSRT-Op3:
    Parameters: [?x0:block, ?x1:block, ?x2:robot]
    Preconditions: [Clear(?x0:block), Gripping(?x2:robot, ?x1:block)]
    Add Effects: [EmptyGripper(?x2:robot), On(?x1:block, ?x0:block)]
    Delete Effects: [Gripping(?x2:robot, ?x1:block)]
    Ignore Effects: []
    Option Spec: Stack(?x2:robot, ?x0:block)
```

### C.2.3 COFFEE

```
1   RobotHoldingJug
2
3   JugTilted
4   ```python
5   def _JugTilted_NSP_holds(state: RawState, objects: Sequence[Object]) -> bool:
6       """Determine if the jug is rotated by a non-zero angle."""
7       jug, = objects
8       # Assuming a rotation value of 0 means upright, any other value implies rotation
9       return abs(state.get(jug, "rot")) > 0.1
10
11  name: str = "JugTilted"
12  param_types: Sequence[Type] = [_jug_type]
13  JugTilted = NSPredicate(name, param_types, _JugTilted_NSP_holds)
14  ```
15
16  JugUpright
17
18  JugInMachine
19  ```python
20  def _JugInMachine_NSP_holds(state: RawState, objects: Sequence[Object]) -> bool:
21      """Jug ?x is placed inside coffee machine ?y."""
22      jug, machine = objects
23      # If the jug is being held, it cannot be in the machine.
24      if _RobotHolding_NSP_holds(state, [state.get_objects(_robot_type)[0], jug]):
25          return False
26
27      # Crop the image to focus on the jug and the coffee machine.
28      attention_image = state.crop_to_objects([jug, machine])
```

```
29      jug_name = jug.id_name
30      machine_name = machine.id_name
31      return state.evaluate_simple_assertion(
32          f"{jug_name} is placed inside {machine_name}.", attention_image
33      )
34
35  name: str = "JugInMachine"
36  param_types: Sequence[Type] = [_jug_type, _machine_type]
37  JugInMachine = NSPredicate(name, param_types, _JugInMachine_NSP_holds)
38  ```
39
40  GripperOpen
```

```
NSRT-Op0:
   Parameters: [?x0:jug, ?x1:robot]
   Preconditions: [GripperOpen(?x1:robot), JugUpright(?x0:jug)]
   Add Effects: [RobotHoldingJug(?x1:robot, ?x0:jug)]
   Delete Effects: [GripperOpen(?x1:robot)]
   Ignore Effects: []
   Option Spec: PickJug(?x1:robot, ?x0:jug)

NSRT-Op1:
   Parameters: [?x0:coffee_machine, ?x1:jug, ?x2:robot]
   Preconditions: [RobotHoldingJug(?x2:robot, ?x1:jug)]
   Add Effects: [GripperOpen(?x2:robot),
   JugInMachine(?x1:jug, ?x0:coffee_machine)]
   Delete Effects: [RobotHoldingJug(?x2:robot, ?x1:jug)]
   Ignore Effects: []
   Option Spec: PlaceJugInMachine(?x2:robot, ?x1:jug,
   ?x0:coffee_machine)

NSRT-Op2:
   Parameters: [?x0:coffee_machine, ?x1:jug, ?x2:robot]
   Preconditions: [JugInMachine(?x1:jug, ?x0:coffee_machine)]
   Add Effects: [JugFilled(?x1:jug)]
   Delete Effects: []
   Ignore Effects: []
   Option Spec: TurnMachineOn(?x2:robot, ?x0:coffee_machine)

NSRT-Op3:
   Parameters: [?x0:coffee_machine, ?x1:jug, ?x2:robot]
   Preconditions: [JugInMachine(?x1:jug, ?x0:coffee_machine)]
   Add Effects: [RobotHoldingJug(?x2:robot, ?x1:jug)]
   Delete Effects: [GripperOpen(?x2:robot),
   JugInMachine(?x1:jug, ?x0:coffee_machine)]
   Ignore Effects: []
   Option Spec: PickJug(?x2:robot, ?x1:jug)

NSRT-Op4:
   Parameters: [?x0:cup, ?x1:jug, ?x2:robot]
   Preconditions: [JugFilled(?x1:jug),
   RobotHoldingJug(?x2:robot, ?x1:jug)]
   Add Effects: [CupFilled(?x0:cup)]
   Delete Effects: [JugFilled(?x1:jug), JugUpright(?x1:jug),
   RobotHoldingJug(?x2:robot, ?x1:jug)]
   Ignore Effects: []
   Option Spec: Pour(?x2:robot, ?x1:jug, ?x0:cup)

NSRT-Op5:
   Parameters: [?x0:jug, ?x1:robot]
   Preconditions: [GripperOpen(?x1:robot)]
   Add Effects: [JugUpright(?x0:jug)]
   Delete Effects: []
   Ignore Effects: []
   Option Spec: Twist(?x1:robot, ?x0:jug)
```

```
NSRT-Op6:
    Parameters: [?x0:coffee_machine, ?x1:jug, ?x2:robot]
    Preconditions: [JugInMachine(?x1:jug, ?x0:coffee_machine)]
    Add Effects: [JugFilled(?x1:jug)]
    Delete Effects: [JugInMachine(?x1:jug, ?x0:coffee_machine)]
    Ignore Effects: []
    Option Spec: TurnMachineOn(?x2:robot, ?x0:coffee_machine)

NSRT-Op7:
    Parameters: [?x0:cup, ?x1:jug, ?x2:robot]
    Preconditions: [JugFilled(?x1:jug),
    RobotHoldingJug(?x2:robot, ?x1:jug)]
    Add Effects: [CupFilled(?x0:cup), JugTilted(?x1:jug)]
    Delete Effects: [JugFilled(?x1:jug),
    RobotHoldingJug(?x2:robot, ?x1:jug)]
    Ignore Effects: []
    Option Spec: Pour(?x2:robot, ?x1:jug, ?x0:cup)

NSRT-Op8:
    Parameters: [?x0:cup, ?x1:jug, ?x2:robot]
    Preconditions: [JugFilled(?x1:jug),
    RobotHoldingJug(?x2:robot, ?x1:jug)]
    Add Effects: [CupFilled(?x0:cup), JugTilted(?x1:jug)]
    Delete Effects: []
    Ignore Effects: []
    Option Spec: Pour(?x2:robot, ?x1:jug, ?x0:cup)

NSRT-Op9:
    Parameters: [?x0:cup, ?x1:jug, ?x2:robot]
    Preconditions: [JugFilled(?x1:jug), RobotHoldingJug(?x2:robot,
    ?x1:jug)]
    Add Effects: [CupFilled(?x0:cup), JugTilted(?x1:jug)]
    Delete Effects: [RobotHoldingJug(?x2:robot, ?x1:jug)]
    Ignore Effects: []
    Option Spec: Pour(?x2:robot, ?x1:jug, ?x0:cup)
```

### C.2.4 COVER HEAVY

```python
EmptyHand
Holding
IsBlack
```python
def _IsBlack_NSP_holds(state: State, objects: Sequence[Object]) -> bool:
    block, = objects
    block_id = block.id_name
    attention_image = state.crop_to_objects([block])
    return state.evaluate_simple_assertion(f"{block_id} is black.", attention_image)

name = "IsBlack"
param_types = [_block_type]
IsBlack = NSPredicate(name, param_types, _IsBlack_NSP_holds)
```
```

```
  NSRT-Op1:
     Parameters: [?x0:block, ?x1:robot, ?x2:target]
     Preconditions: [Holding(?x1:robot, ?x0:block)]
     Add Effects: [Covers(?x0:block, ?x2:target), EmptyHand(?x1:robot)]
     Delete Effects: [Holding(?x1:robot, ?x0:block)]
     Ignore Effects: []
     Option Spec: Place(?x0:block, ?x2:target)
  NSRT-Op0:
     Parameters: [?x0:block, ?x1:robot]
     Preconditions: [IsBlack(?x0:block), EmptyHand(?x1:robot)]
     Add Effects: [Holding(?x1:robot, ?x0:block)]
     Delete Effects: [EmptyHand(?x1:robot)]
     Ignore Effects: []
     Option Spec: Pick(?x0:block)
```

### C.2.5 BALANCE

```
1   OnPlate
2   ```
3   def _OnPlate_CP_holds(atoms: Set[GroundAtom],
4                             objects: Sequence[Object]) -> bool:
5     x, y = objects
6     for atom in atoms:
7        if atom.predicate == DirectlyOnPlate and\
8              atom.objects == [x, y]:
9           return True
10    other_blocks = {a.objects[0] for a in atoms if
11                          a.predicate == DirectlyOn or\
12                          a.predicate == DirectlyOnPlate}
13
14    for other_block in other_blocks:
15       holds1 = False
16       for atom in atoms:
17          if atom.predicate == DirectlyOn and\
18                atom.objects == [x, other_block]:
19             holds1 = True
20             break
21       if holds1 and _OnPlate_CP_holds(atoms, [other_block, y]):
22          return True
23    return False
24
25  name: str = "OnPlate"
26  param_types: Sequence[Type] = [_block_type, _plate_type]
27  OnPlate = ConceptPredicate(name, param_types, _OnPlate_CP_holds)
28  ```
29
30  BlocksDistributedEvenly
31  ```
32  def _BlocksDistributedEvenly_CP_holds(atoms: Set[GroundAtom],
33             objects: Sequence[Object]) -> bool:
34    plate1, plate2 = objects
35    if plate1 == plate2:
36       return False
37    count1 = 0
38    count2 = 0
39    for atom in atoms:
40       if atom.predicate == OnPlate:
41          if atom.objects[1] == plate1:
42             count1 += 1
43          elif atom.objects[1] == plate2:
44             count2 += 1
45    return count1 == count2
46
47  name: str = "BlocksDistributedEvenly"
48  param_types: Sequence[Type] = [_plate_type, _plate_type]
49  BlocksDistributedEvenly = ConceptPredicate(name, param_types,
50          _BlocksDistributedEvenly_CP_holds)
51  ```
```

```
NSRT-Unstack:
   Parameters: [?block:block, ?otherblock:block, ?robot:robot]
   Preconditions: [Clear(?block:block), DirectlyOn(?block:block,
   ?otherblock:block), GripperOpen(?robot:robot)]
   Add Effects: [Clear(?otherblock:block), Holding(?block:block)]
   Delete Effects: [Clear(?block:block), DirectlyOn(?block:block,
   ?otherblock:block), GripperOpen(?robot:robot)]
   Ignore Effects: []
   Option Spec: Pick(?robot:robot, ?block:block)
NSRT-Op3:
   Parameters: [?block:block, ?otherblock:block, ?robot:robot]
   Preconditions: [Clear(?otherblock:block), Holding(?block:block)]
   Add Effects: [Clear(?block:block), DirectlyOn(?block:block,
   ?otherblock:block), GripperOpen(?robot:robot)]
   Delete Effects: [Clear(?otherblock:block), Holding(?block:block)]
   Ignore Effects: []
   Option Spec: Stack(?robot:robot, ?otherblock:block)
NSRT-Op2:
   Parameters: [?x0:machine, ?x1:plate, ?x2:plate, ?x3:robot]
   Preconditions: [BlocksDistributedEvenly(?x2:plate, ?x1:plate)]
   Add Effects: [MachineOn(?x0:machine)]
   Delete Effects: []
   Ignore Effects: []
   Option Spec: TurnMachineOn(?x3:robot, ?x1:plate, ?x2:plate)
NSRT-Op4:
   Parameters: [?block:block, ?robot:robot, ?plate:plate]
   Preconditions: [ClearPlate(?plate:plate), Holding(?block:block)]
   Add Effects: [Clear(?block:block), DirectlyOnPlate(?block:block,
   ?plate:plate), GripperOpen(?robot:robot)]
   Delete Effects: [ClearPlate(?plate:plate), Holding(?block:block)]
   Ignore Effects: []
   Option Spec: PutOnPlate(?robot:robot, ?plate:plate)
NSRT-PickFromTable:
   Parameters: [?block:block, ?robot:robot, ?plate:plate]
   Preconditions: [Clear(?block:block), DirectlyOnPlate(?block:block,
   ?plate:plate), GripperOpen(?robot:robot)]
   Add Effects: [Holding(?block:block)]
   Delete Effects: [Clear(?block:block), DirectlyOnPlate(?block:block,
   ?plate:plate), GripperOpen(?robot:robot)]
   Ignore Effects: []
   Option Spec: Pick(?robot:robot, ?block:block)
```

## C.3 FURTHER PLANNING STATISTICS

The average planning node expanded and wall-time statistics for our approach, alongside other planning approaches, are summarized in Table 1.

In the Blocks and Balance domains, our use of derived predicates is not out-of-box compatible with relaxed planning heuristics, such as LM-cut, which we typically employ through Pyperplan. As a result, we resorted to a simpler goal-count heuristic, which estimates the distance to the goal by counting the number of unsatisfied goals. This heuristic is less informed than LM-cut, leading to significantly larger node expansions and longer planning times in these domains than expected. In future work, we aim to develop a version of LM-cut that is compatible with derived *NSPs*.

## D ADDITIONAL ENVIRONMENT DETAILS

**Cover.** This environment has goal predicate {Covers(?x:block, ?y:target)}. The initial operators are:

| Environment | Ours | | | Oracle | | | Sym. pred. | | |
|---|---|---|---|---|---|---|---|---|---|
| | Succ | Node | Time | Succ | Node | Time | Succ | Node | Time |
| Cover | 100.0 | 9.4 | 0.142 | 100.0 | 8.4 | 0.129 | 100.0 | 26.9 | 0.151 |
| Blocks | 96.0 | 1117675 | 254.621 | 94.0 | 550630 | 101.737 | 7.2 | 121.4 | 4.279 |
| Cover Heavy | 97.0 | 7.9 | 0.057 | 100.0 | 5.4 | 0.060 | 46.0 | 5.7 | 0.061 |
| Coffee | 65.3 | 40.3 | 0.969 | 99.3 | 19.3 | 0.652 | 68.0 | 199.4 | 3.270 |
| Balance | 100.0 | 26.3 | 0.856 | 100.0 | 30.6 | 0.585 | 20.0 | 12.2 | 0.125 |

| Environment | Ours | | | Ablate op. | | | No invent | | |
|---|---|---|---|---|---|---|---|---|---|
| | Succ | Node | Time | Succ | Node | Time | Succ | Node | Time |
| Cover | 100.0 | 9.4 | 0.142 | 100.0 | 7.0 | 0.148 | 68.0 | 28.1 | 0.113 |
| Blocks | 96.0 | 1117675 | 254.621 | 12.0 | 24.8 | 0.222 | 1.3 | 321.0 | 0.224 |
| Cover Heavy | 97.0 | 7.9 | 0.057 | 46.0 | 5.7 | 0.128 | 36.7 | 29.5 | 0.099 |
| Coffee | 65.3 | 40.3 | 0.969 | 65.3 | 29.6 | 2.441 | 0.0 | – | – |
| Balance | 100.0 | 26.3 | 0.856 | 100.0 | 28.0 | 1.180 | 25.3 | 13.5 | 0.204 |

Table 1: Further planning statistics.

```
NSRT-Pick:
    Parameters: [?block:block]
    Preconditions: []
    Add Effects: []
    Delete Effects: []
    Ignore Effects: []
    Option Spec: Pick(?block:block)

NSRT-Place:
    Parameters: [?block:block, ?target:target]
    Preconditions: []
    Add Effects: [Covers(?block:block, ?target:target)]
    Delete Effects: []
    Ignore Effects: []
    Option Spec: Place(?block:block, ?target:target)
```

**Blocks.** This environment has goal predicates: {On(?x:block, ?y:block), OnTable(?x:block)} and initial operators

```
NSRT-PickFromTable:
   Parameters: [?block:block, ?robot:robot]
   Preconditions: []
   Add Effects: []
   Delete Effects: [OnTable(?block:block)]
   Ignore Effects: []
   Option Spec: Pick(?robot:robot, ?block:block)

NSRT-PutOnTable:
  Parameters: [?block:block, ?robot:robot]
  Preconditions: []
  Add Effects: [OnTable(?block:block)]
  Delete Effects: []
  Ignore Effects: []
  Option Spec: PutOnTable(?robot:robot)

NSRT-Stack:
  Parameters: [?block:block, ?otherblock:block,
  ?robot:robot]
  Preconditions: []
  Add Effects: [On(?block:block,
  ?otherblock:block)]
  Delete Effects: []
  Ignore Effects: []
  Option Spec: Stack(?robot:robot,
  ?otherblock:block)

NSRT-Unstack:
  Parameters: [?block:block, ?otherblock:block,
  ?robot:robot]
  Preconditions: []
  Add Effects: []
  Delete Effects: [On(?block:block,
  ?otherblock:block)]
  Ignore Effects: []
  Option Spec: Pick(?robot:robot, ?block:block)
```

**Coffee.** This environment has goal predicates: {CupFilled(?cup:cup)}. We include the predicate JugFilled(?jug:jug) in the initial set of predicates because it was very challenging to have a VLM to determine this especially with the graphics in the simulator. It has initial operators:

```
NSRT-PickJugFromTable:
   Parameters: [?robot:robot, ?jug:jug]
   Preconditions: []
   Add Effects: []
   Delete Effects: []
   Ignore Effects: []
   Option Spec: PickJug(?robot:robot, ?jug:jug)

NSRT-PlaceJugInMachine:
   Parameters: [?robot:robot, ?jug:jug,
   ?machine:coffee_machine]
   Preconditions: []
   Add Effects: []
   Delete Effects: []
   Ignore Effects: []
   Option Spec: PlaceJugInMachine(?robot:robot,
       ?jug:jug, ?machine:coffee_machine)

NSRT-PourFromNowhere:
   Parameters: [?robot:robot, ?jug:jug,
   ?cup:cup]
   Preconditions: []
   Add Effects: [CupFilled(?cup:cup)]
   Delete Effects: []
   Ignore Effects: []
   Option Spec: Pour(?robot:robot, ?jug:jug,
   ?cup:cup),

NSRT-TurnMachineOn:
   Parameters: [?robot:robot, ?jug:jug,
   ?machine:coffee_machine]
   Preconditions: []
   Add Effects: [JugFilled(?jug:jug)]
   Delete Effects: []
   Ignore Effects: []
   Option Spec: TurnMachineOn(?robot:robot,
       ?machine:coffee_machine),

NSRT-Twist:
   Parameters: [?robot:robot, ?jug:jug]
   Preconditions: []
   Add Effects: []
   Delete Effects: []
   Ignore Effects: []
   Option Spec: Twist(?robot:robot, ?jug:jug)
```

**Cover Heavy.** This has the same set of goal predicates and operators as Cover.

**Balance.** This has the goal predicate: {MachineOn(?x:machine)}. Here we consider a continual learning setting where the agent is initialized with the abstractions commonly found in Blocks. They are {Clear(?x:block), ClearPlate(?x:plate), DirectlyOn(?x:block, ?y:block), DirectlyOnPlate(?x:block, ?y:plate), GripperOpen(?x:robot), Holding(?x:block)}. The initial set of operators is:

```
NSRT-PickFromTable:
   Parameters: [?block:block, ?robot:robot,
   ?plate:plate]
   Preconditions: [Clear(?block:block),
   DirectlyOnPlate(?block:block, ?plate:plate),
   GripperOpen(?robot:robot)]
   Add Effects: [Holding(?block:block)]
   Delete Effects: [Clear(?block:block),
   DirectlyOnPlate(?block:block, ?plate:plate),
   GripperOpen(?robot:robot)]
   Ignore Effects: []
   Option Spec: Pick(?robot:robot, ?block:block)

NSRT-PutOnPlate:
   Parameters: [?block:block, ?robot:robot,
   ?plate:plate]
   Preconditions: [ClearPlate(?plate:plate),
   Holding(?block:block)]
   Add Effects: [Clear(?block:block),
   DirectlyOnPlate(?block:block, ?plate:plate),
   GripperOpen(?robot:robot)]
   Delete Effects: [ClearPlate(?plate:plate),
   Holding(?block:block)]
   Ignore Effects: []
   Option Spec: PutOnPlate(?robot:robot, ?plate:plate),

NSRT-Stack:
   Parameters: [?block:block, ?otherblock:block,
   ?robot:robot]
   Preconditions: [Clear(?otherblock:block),
   Holding(?block:block)]
   Add Effects: [Clear(?block:block),
   DirectlyOn(?block:block, ?otherblock:block),
   GripperOpen(?robot:robot)]
   Delete Effects: [Clear(?otherblock:block),
   Holding(?block:block)]
   Ignore Effects: []
   Option Spec: Stack(?robot:robot,
    ?otherblock:block)

NSRT-Unstack:
   Parameters: [?block:block, ?otherblock:block,
   ?robot:robot]
   Preconditions: [Clear(?block:block),
   DirectlyOn(?block:block, ?otherblock:block),
   GripperOpen(?robot:robot)]
   Add Effects: [Clear(?otherblock:block),
   Holding(?block:block)]
   Delete Effects: [Clear(?block:block),
   DirectlyOn(?block:block, ?otherblock:block),
   GripperOpen(?robot:robot)]
   Ignore Effects: []
   Option Spec: Pick(?robot:robot,
   ?block:block)

NSRT-TurnMachineOn:
   Parameters: [?robot:robot, ?machine:machine,
   ?plate1:plate, ?plate2:plate]
   Preconditions: []
   Add Effects: [MachineOn(?machine:machine)]
   Delete Effects: []
   Ignore Effects: []
   Option Spec: TurnMachineOn(?robot:robot,
   ?plate1:plate, ?plate2:plate)
```

