# OpenReview forum: "VisualPredicator: Learning Abstract World Models with Neuro-Symbolic Predicates for Robot Planning"
_ICLR.cc/2025/Conference — ICLR 2025 Spotlight_

### Official Review · Reviewer_JR1Z · 2024-10-27

**Soundness:** 3
**Presentation:** 3
**Contribution:** 2
**Rating:** 6
**Confidence:** 2

**Summary:**

This paper proposes a novel framework for robot planning by integrating NSPs, a method that combines neural and symbolic representations. The authors present an algorithm for online predicate invention, leveraging VLMs to dynamically learn abstract world models through real-time interaction in various robotic environments. The study compares the proposed approach to several HRL and symbolic methods, evaluating performance in terms of sample efficiency, generalization, and interpretability.

**Strengths:**

1.	**Innovative Approach**: The paper introduces neuro-symbolic predicates to robot planning, using VLMs to invent predicates in a task-relevant manner, a strategy that has clear advantages over static symbolic approaches. The approach also supports perceptual abstraction and logical complexity, enabling the system to better generalize across environments.
2.	**Experimental Design**: The authors evaluate their approach across five different simulated robotic domains with various tasks, making the results generalizable. Furthermore, the comparisons with other SOTA, are well-chosen and highlight the advantages of the proposed method in terms of efficiency and adaptability.
3.	**Methodology and Algorithmic Design**: The online NSP invention algorithm is well-structured and incrementally builds task-relevant abstractions. By leveraging both VLMs for perceptual understanding and a symbolic component for logical manipulations, this approach improves sample efficiency, crucial in robotic applications where data is limited or expensive to obtain.

**Weaknesses:**

1.	**Clarity of Explanation**: Some explanations in the paper are unclear. For example, in the paragraph beginning with line 178, the process of conditioning on the previous action should be explained in more detail. Additional details on how these predicate proposals are validated would improve the comprehensibility of the approach.
2.	**Reliance on VLMs and Practicality**: A potential limitation of this approach is its reliance on the accuracy of VLMs, which may be sensitive to changes in the environment. The approach assumes that perceptual cues relevant to task abstractions are identifiable through VLMs. However, variability in VLM performance, particularly with domain shifts or challenging visual inputs, can affect the system's robustness. Although the authors attempt to address this by conditioning predicates on past actions and object labeling, it is unclear how effective these strategies will be in practice, especially under less predictable real-world variations where perceptual noise could affect predicate accuracy.
3.	**Time Complexity and Performance**: As an online learning method, computational efficiency is critical, particularly for real-time applications in robotics. While the paper presents experimental results in supplementary materials, these do not convincingly demonstrate that the proposed approach maintains efficiency at scale. Including more detailed time complexity analysis or exploring algorithmic optimizations could address this concern, as current evaluations suggest that scalability in time-sensitive or resource-constrained environments may be limited. Moreover, the supplementary results lack a detailed breakdown, which would clarify how computational resources scale with task complexity.

**Questions:**

1. How does the framework ensure a robust and diverse predicate set while maintaining task relevance?
2. Is there a systematic mechanism for leveraging failures to refine or adjust future planning decisions, rather than simply discarding infeasible options?

---

> ### Author Response · Authors · 2024-11-21
>
> Thank you for your detailed review and constructive feedback. We address your concerns point by point below:
>
> **Clarity of Explanation.** You highlighted the need for more detailed explanations of key processes, particularly how predicates condition on previous actions. We have addressed this by:
> * Adding an annotated example to Appendix B.1 that demonstrates how prior actions, observations, and truth values are leveraged in the predicate evaluation.
> * Expanding Section 5.3 with a clearer explanation of how the predicates are validated.
>
> **Reliance on VLMs and Practicality.** We agree that the reliance on VLMs introduces potential variability and would like to highlight that our system is designed to handle this in two additional ways:
> 1. Neuro-symbolic reasoning: Our method does not exclusively rely on VLMs. Instead, we combine symbolic reasoning with VLM question answering to combine the strength of both processes.
> 2. Iterative Optimization: The iterative predicate proposal-and-selection process allows the agent to refine its abstraction implementation, optimizing for robustness and reliability.
> We see further improvement in this as an exciting area for future research and have added Appendix B.5 to discuss this. This section outlines the system’s current limitations to handle variability and suggests future extensions.
>
> **Time Complexity and Performance.** You rightly emphasized the importance of computational efficiency, particularly for online learning in robotics.
> Our approach is efficient because each predicate invention iteration completes within minutes on a single GPU, and once abstractions are learned, planning is extremely fast due to the use of domain-independent heuristics and symbolic planners from the AI planning community.
> To further clarify, we have included a more detailed time complexity analysis in Section 5.3 and Appendix B.2 and C.3, illustrating the scalability of our method across increasing task complexities. We believe this demonstrates the approach’s feasibility for real-world applications.
>
> **Robust, diverse yet task-relevant predicate set.** Diversity and relevance are key characteristics of candidate predicates, and our method explicitly promotes these qualities:
> On one hand, we propose predicates using three complementary strategies (described in Section 5.2 and detailed in Appendix B.4): discrimination, transition modeling, and logical derivation.
> On the other hand, our predicate selection process balances task relevance and simplicity through the two predicate score functions. Section 5.3 has been revised for clarity on this process.
>
> **Mechanism for leveraging failure.** We can think of two orthogonal ways to leverage infeasible plans to inform future planning decisions.
> The first is to learn from failed plans by inventing predicates and learning the operators, and the second is to improve the planner used. Our approach is an instance of the first one, and we see the second as an interesting direction for future research.
> Specifically, our approach invents predicates (in particular with the prompting strategy #1) and learns operators to prevent the controllers that failed in previous planning attempts from being selected again in similar states.
> On the other hand, within the planner--specifically for the A* search--we can try to learn a node expansion policy, where the policy could decide how to expand the existing partial plans at the current state, based on the knowledge of previous infeasible plans.
>
> In summary, we have revised the manuscript to address your concerns and to better articulate the contributions of our work. We hope these improvements clarify the robustness and practicality of our approach. Thank you again for your thoughtful review and for helping us refine this work!

---

> > ### Comment · Reviewer_JR1Z · 2024-11-22
> >
> > Thanks to the authors for their detailed response and further analysis. My concerns have been addressed, and I will improve my score accordingly.

---

### Official Review · Reviewer_qh55 · 2024-11-04

**Soundness:** 3
**Presentation:** 3
**Contribution:** 3
**Rating:** 8
**Confidence:** 5

**Summary:**

This paper presents a new method to learn grounded symbolic predicates and operators from online interaction data for high-level planning. The core of this method is to generate predicates represented in python functions by prompting an VLM. Such predicate functions can invoke VLMs to process image observations and incorporate logical structure to build predicate hierarchies (i.e., derived predicates). Then the most useful predicates are selected and operators are learned following variants of previous works [1, 2]. In experiments, the method is evaluated on 5 domains with multiple baseline methods including Hierarchical RL and VLM planning, showingcase that the method can recover symbolic planning domains that allows effective planning.

[1] Silver, Tom, et al. "Predicate invention for bilevel planning." Proceedings of the AAAI Conference on Artificial Intelligence. Vol. 37. No. 10. 2023.

[2] Chitnis, Rohan, et al. "Learning neuro-symbolic relational transition models for bilevel planning. In 2022 IEEE." RSJ International Conference on Intelligent Robots and Systems (IROS).

**Strengths:**

This paper has sufficient quality and soundness, and I'm satisfied with its approach and evaluations.
- The paper is well-structured and clearly-presented. It has good coverage of literature.
- Learning symbolic predicates is foundamental to high-level planning. The paper proposes a flexible approach to learn neural-symbolic predicates grounded on image observations and natural language.
- The proposed method is comprehensively evaluated via comparison with baselines and ablation studies.

**Weaknesses:**

- I would suppose the VLM-based predicate functions can get wrong even after applying the techniques described in line 178-189. How will the invented predicates with wrong grounding affect downstream operator learning and planning?
- I wonder how are the low-level skills realized in experiments for the proposed method and baselines. Are they realized by motion planning? Do you learn low-level skills in the MAPLE baseline?
- In Figure 4, it seems that the oracle baseline doesn't achieve 100% success rate in Blocks and Coffee domain. Its performance is even slightly worse than the proposed method in Blocks. This seems to be weird as the Blocks domain is the most well-known classical domain that can be solved completely with a few predicates and operators. I wonder why this happens?

**Questions:**

See Weakness

---

> ### Author Response · Authors · 2024-11-21
>
> Thank you for your positive assessment and valuable comments. Below, we address each of your concerns in detail.
>
> **VLM reliability.** We agree that reliance on VLMs can introduce variability, and addressing this issue has been a key consideration in our design. Our system mitigates this through:
> 1. Neuro-symbolic reasoning: Our method does not exclusively rely on VLMs. Instead, we complement them with deterministic symbolic computations, enhancing generalization and improving the system’s overall robustness.
>
> 2. Iterative Optimization: The iterative predicate proposal-and-selection process allows the agent to refine its abstraction implementation over time, optimizing for robustness and reliability.
> We acknowledge this as an open challenge and an exciting area for future research. We have added a discussion in Appendix B.5, where we outline the limitations and propose potential extensions.
>
> **Clarification for low-level skills.** We design hand-coded closed-loop controllers for each environment and these are used across all the approaches.
> We have also clarified this under the environment paragraph of Section 6.
>
> **Oracle performance.** Thank you for noting the anomaly in the Oracle baseline’s performance in the Blocks. We identified this is due to the suboptimal implementation of the `place` controller. We have fixed this issue and updated the results accordingly in Figure 4 and the corresponding sections of the manuscript.

---

> > ### Comment · Reviewer_qh55 · 2024-11-25
> >
> > Thanks to the authors for the explanation. I like this paper, and I will keep my score.

---

### Official Review · Reviewer_oEZx · 2024-11-04

**Soundness:** 4
**Presentation:** 4
**Contribution:** 3
**Rating:** 8
**Confidence:** 3

**Summary:**

This paper introduces Neuro-Symbolic Predicates (NSPs), a first-order abstraction language that combines symbolic and neural knowledge representations for high-level planning in robotics. The authors propose an online algorithm for inventing such predicates and learning abstract world models from interactions with the environment, without relying on demonstrations. The method leverages Vision-Language Models (VLMs) to ground predicates in perceptual data and integrates them into a planning framework that interleaves predicate proposal, validation, and goal-driven exploration. The approach is evaluated across five simulated robotic domains, comparing it with hierarchical reinforcement learning, VLM planning, and symbolic predicate invention methods. The results demonstrate that the proposed method offers better sample efficiency, stronger out-of-distribution generalization, and improved interpretability.

**Strengths:**

1. Novel Approach: This paper proposes a novel approach to effectively integrate symbolic reasoning with neural perceptual representations using NSPs to address the challenge of forming task-specific abstractions in robotics.

2. Online Predicate Invention: The proposed method uses online invention of predicates from environment interactions that does not need demonstrations. This is a valuable property in real-world applications, since collecting demonstrations could be expensive or even infeasible.

3. Integration of VLMs: VLMs are leveraged to enable the agent to handle rich representations of concepts, both visually and logically, grounding predicates in the perceptual data, thus enabling more flexible and stronger abstractions.

4. Strong Empirical Results: Their results show strong empirical results, outperforming other baselines over various domains in terms of sample efficiency and generalization to unseen tasks, and planning efficiency.

5. Interpretability: The model uses NSPs, providing interpretable outputs. This helps in understanding or debugging the behavior of the agent. This is critical in safety-critical applications.

**Weaknesses:**

1. The reliance on VLMs raises questions about its scalability to more complex or noisy real-world settings in which VLMs might perform poorly, hence affecting the robustness of the method.

2. In their approach it is assumed that there is access beforehand to well-defined motor skills and object-centric state representations that in practice are not always available.

3. Comparisons to more SOTA methods in hierarchical planning or neuro-symbolic reasoning could strengthen empirical evaluations and position the work better within the literature.

**Questions:**

1. How would your method scale to real-world environments with more complex visual scenes and possible noise? Have you considered the potential limitation of current VLMs in these settings, and what could be the effect on NSPs' performance?

2. Considering the fact that VLMs are not infallible, how sensitive is your approach to inaccuracies or biases in the VLM outputs? Do you observe cases where VLM errors lead to incorrect predicate invention or planning failures, and how could these be mitigated?

3. Your approach assumes a given fixed set of low-level motor skills. How much can that assumption be relaxed, and could the approach be extended towards learning or adapting these skills jointly with the high-level predicates?

---

> ### Author Response · Authors · 2024-11-21
>
> Thank you for your thoughtful feedback. We appreciate your positive comments and have carefully addressed each of your concerns and questions below.
> Regarding the weaknesses identified by you, for example:
> >  Have you considered applying D-TSN to real-world tasks or environments with higher-dimensional observation spaces, such as language or vision?
>
> We think that it's possible the weaknesses section of your review was meant for another paper you might be reviewing (about [D-TSN](https://openreview.net/forum?id=v593OaNePQ), not NSP) which doesn't involve vision (as we learn from images). We suspect this was an innocent copy-paste mistake. We're really happy to have your support, and we'd welcome discussion regarding limitations of our paper.
>
> **Observation Noise and VLM Reliability.** To address questions 1 and 2 together, we fully agree that the reliability of visual language models (VLMs) under noisy observations could pose challenges for our approach.
> However, we have designed the system to handle this in the following 2 ways:
> 1. Neuro-symbolic reasoning: Our method does not solely rely on VLMs. Instead, we combine VLM outputs with symbolic reasoning, improving both robustness and flexibility.
> 2. Iterative Optimization: The iterative predicate proposal-and-selection process allows the agent to refine its abstraction implementation, optimizing for robustness and reliability.
> We recognize that further improvements in this area are crucial and have added a discussion of these limitations and future directions in Appendix B.5. This section outlines the current challenges and suggests extensions for handling observation noise and VLM inaccuracies more effectively.
>
> **Assumption about skills.** We agree that relaxing the assumption of fixed low-level motor skills is a fascinating direction for future research. Learning skills can leverage advances in closed-loop policy writing (e.g., Code as Policies), reinforcement learning for manipulation skills (e.g., RMA$^2$, Eureka), and motion planning (e.g., ReKep).
> Additionally, the interplay between symbols and skills is an exciting area for exploration. For example, predicates can serve as subgoals for controllers or as components of reward functions for learning new skills.
> We see this virtuous cycle as a promising avenue for future work.
> We have included some discussion of this in the revised manuscript.

---

> > ### Comment · Reviewer_oEZx · 2024-11-25
> >
> > Thank you for addressing the areas of concern and reflecting them in your responses and the updated revision.
> >
> > I apologize for the confusion caused by the weaknesses section in my initial review and for any inconvenience this may have caused. I believe that I mistakenly included the wrong review from my notes when submitting the final versions. I have now updated the review to replace the weaknesses section with the one I had intended to submit. That said, the points raised remain related to the questions and concerns you have already addressed in your response.
> >
> > I am maintaining the same overall score for your submission. Best of luck with your paper!

---

### Author Response · Authors · 2024-11-21

We are grateful for the reviewers’ detailed assessments and the recognition of the contributions of our work. We are encouraged by the remark that the work is "a novel and innovative approach to effectively integrate symbolic reasoning with neural representations", has "strong empirical results", is "comprehensively evaluated", and is "well-structured and clearly presented".

In response to the reviewers’ constructive feedback, we have made targeted improvements to the manuscript, mainly including:
* Enhanced clarity in the explanation of key processes, such as predicate conditioning on previous actions, with added examples and expanded discussion in Sections 5.3 and Appendix B.1.
* Acknowledged the unreliability in VLM performance and highlighted mechanisms, such as neuro-symbolic reasoning and iterative optimization, to mitigate this. We have also added Appendix B.5 to outline limitations and propose extensions.
* Provided additional computational complexity analysis and planning statistics in Sections 5.3, Appendix B.2, and C.3, addressing concerns about scalability and real-world feasibility.
* Clarified the predicate selection process and the mechanisms for balancing diversity and task relevance, with additional detail in Section 5.3 and Appendix B.4.
* Discussed leveraging failure mechanisms, both through predicate invention and potential improvements to planner policies, as future directions.

To further improve the quality of our work, we have
* increased the number of random seeds in the experiments from 3 to 5 and updated the results;
* added Appendix C.1 to include an example online learning trajectory illustrating the progressive predicate invention process and performance improvement.

We appreciate the reviewers’ thoughtful critiques and believe the response and revised manuscript better articulates the contributions, robustness, and practical value of our approach.

---

### Meta-Review · Area_Chair_qm8H · 2024-12-24

**Metareview:**

The paper presents a method for predicate learning based on VLMs. The reviewers highlighted the importance of the problem, the novel use of VLMs, and the strong empirical results. Reviewers noted several weaknesses, primarily around robustness of VLMs and scalability, which were addressed by the authors. Overall, the reviewers universally recommended acceptance.

**Additional Comments On Reviewer Discussion:**

All reviewers highlighted robustness and scalability of VLMs as a potentially major issue. The authors highlighted to all reviewers that the combination of neuro-symbolic reasoning and iterative optimization yields greater robustness. While the authors could have improved experimentation on these questions, the reviewers are satisfied with this approach.

Several reviewers mentioned the fixed low-level skills used. The authors clarified that they were hand-designed, but work on these is beyond the scope of the paper which is a reasonable boundary to draw.

Finally, there was discussion on the computational complexity and scalability of the method, but the answers provided by the authors were satisfactory. The authors also added an analysis of complexity to the paper.

---

### Decision · Program_Chairs · 2025-01-22

Accept (Spotlight)